# Genetic Dissection of Panicle Morphology Traits in Super High-Yield Hybrid Rice Chaoyou 1000

**DOI:** 10.3390/plants13020179

**Published:** 2024-01-09

**Authors:** Jing Jiang, Li Wang, Gucheng Fan, Yu Long, Xueli Lu, Run Wang, Haiyang Liu, Xianjin Qiu, Dali Zeng, Zhixin Li

**Affiliations:** 1Institute of Crop Genetics and Breeding, Yangtze University, Jingzhou 434025, China; j1579454048@163.com (J.J.); hyliu@yangtzeu.edu.cn (H.L.);; 2China National Rice Research Institute, Hangzhou 310006, Chinadalizeng@126.com (D.Z.)

**Keywords:** QTLs, rice, genetic map, morphology trait of panicle

## Abstract

The morphological characteristics of the rice panicle play a pivotal role in influencing yield. In our research, we employed F_2_ and F_2:3_ populations derived from the high-yielding hybrid rice variety Chaoyou 1000. We screened 123 pairs of molecular markers, which were available, to construct the genetic linkage map. Subsequently, we assessed the panicle morphology traits of F_2_ populations in Lingshui County, Hainan Province, in 2017, and F_2:3_ populations in Hangzhou City, Zhejiang Province, in 2018. These two locations represent two types of ecology. Hangzhou’s climate is characterized by high temperatures and humidity, while Lingshui’s climate is characterized by a tropical monsoon climate. In total, 33 QTLs were identified, with eight of these being newly discovered, and two of them were consistently detected in two distinct environments. We identified fourteen QTL-by-environment interactions (QEs), which collectively explained 4.93% to 59.95% of the phenotypic variation. While most of the detected QTLs are consistent with the results of previous tests, the novel-detected QTLs will lay the foundation for rice yield increase and molecular breeding.

## 1. Introduction

Thus far, agricultural production has managed to keep pace with the rapidly expanding global population. However, it is projected that by 2050, crop production will face a shortage [1,2]. The global demand for staple crop products is expected to surge by 60% from 2010 to 2050 [2]. Rice (*Oryza sativa* L.) is one of the most important crops in the world. It serves as a primary source of sustenance for human life and has been cultivated for nearly 10,000 years [3]. This cereal crop plays a vital role in human development and social progress. In 1986 and 1996, China, in collaboration with the International Rice Research Institute, initiated the Super Rice Breeding Program as a staged effort to augment rice yield [4]. In 2005 and 2007, Zhang Qifa twice introduced the concept of Green Super Rice [5,6]. Following a decade of unwavering dedication by plant breeders, China has identified a collection of outstanding green super rice varieties. The cumulative promotion area of these varieties in the country has now reached 6 million hectares. Hence, increasing rice yield is still the primary goal of breeders.

The panicle serves as the principal storage organ for photosynthetic products in rice. Furthermore, it also can conduct photosynthesis. Optimal panicle morphology is essential for the synthesis, transport, and storage of photosynthetic products, which is crucial for enhancing rice yield. The characteristics of the rice panicle represent a complex quantitative trait and play a pivotal role in determining yield [7,8]. Panicle-related traits are influenced by both genetic components and environmental factors [9]. In the past decades, with the development of molecular marker technology and genetic statistical models, more and more panicle-related trait QTLs have been mapped and cloned. To date, numerous panicle-related genes have been successfully mapped and, in some cases, even cloned [10,11]. For example, genes like *Gn1a* [12], *EP3* [13], *qSpp-2* [14], *qSpp-3* [15], *qGN 4-1* [16], *Spr3* [17], *Ghd7* [18], and more.

While we have made significant progress in identifying key QTLs and genes related to rice panicles, our comprehension of the gene networks governing rice panicle development remains limited. Hence, our foremost objective is to continue uncovering additional panicle QTLs as we strive to increase rice yields [19]. Exceptional panicle morphology traits are crucial for achieving a super high yield of rice. Hence, it is essential to detect novel QTLs from super hybrid rice to achieve a high yield of rice. In this study, we utilized a genetic map constructed by the F_2_ and F_2:3_ population of Chaoyou 1000, a super high-yielding hybrid rice variety, to detect QTLs associated with nine panicle morphology traits. We analyzed epistatic interactions in QTLs, and their interactions with the environment, and identified clusters of QTLs with multiple effects in two distinct environmental conditions. Our primary aim is to identify novel QTLs associated with panicle traits from the super hybrid rice variety.

## 2. Result

### 2.1. Phenotypic Analysis of Parents and Populations

The panicle morphology traits of their parents and two populations were demonstrated in Figure 1 and Figure 2 and Appendix A. Notably, all panicle-related traits exhibited significant differences in both Lingshui and Hangzhou, except for empty grain number per panicle (EGN) and seed setting rate (SSR) in Lingshui. When compared to R900, Guangxiang 24S exhibited a higher panicle number per plant (PN) in both environments and a greater empty grain number per panicle (EGN) in Hangzhou, but all other traits, including panicle length (PL), primary branch number (PBN), secondary branch number (SBN), grain number per panicle (GN), filled grain number per panicle (FGN), seed setting rate (SSR), and panicle setting density (PSD), were significantly lower than R900 in both environments (Figure 1). Guangxiang 24S demonstrates a greater sensitivity to the environment, as reflected in substantial variations in the values of PN, PL, SBN, GN, FGN, EGN, and SSR between Lingshui and Hangzhou. In contrast, R900 exhibits significant differences only in the values of EGN between the two locations, highlighting a distinct disparity in environmental sensitivity between the two varieties.

In the F_2_ and F_2:3_ populations, significant segregation is observed in PN, PL, EGN, and SSR, suggesting notable environmental sensitivity in these traits. Furthermore, there is notable variation in PL, SBN, GN, EGN, and SSR, indicating substantial differences in the associated loci between the two parental varieties. All the panicle morphology traits of the two populations displayed continuous distributions (Appendix A, Figure 2), indicating they were typical quantitative traits, and suitable for QTL mapping.

### 2.2. Correlation Analysis of Panicle-Related Traits

The correlation analysis between the panicle traits of the two populations is presented in Figure 3. Except for PN in the F_2:3_ population, GN exhibited a significantly positive correlation with PN, PL, PBN, and SBN in both populations. This suggests that plants with longer PL, more PBN, and SBN tend to have a greater number of GN. In both the F_2_ and F_2:3_ populations, SBN and GN were significantly positively correlated, with a notably higher correlation coefficient. Over two years, GN showed a strong positive correlation with both PBN and SBN, particularly with SBN, ultimately resulting in a high PSD. In addition to being positively correlated with FGN and negatively correlated with EGN, SSR is also negatively correlated with PL, suggesting that an excessively long panicle length may lead to a reduction in seed setting rate.

Given the substantial differences in the correlation of panicle traits between the two populations in the two environments, an analysis was conducted to examine the correlation between the same traits in the two populations. The findings revealed that there was no significant correlation between PN, EGN, and SSR for panicle traits in the two environments. Because the F_2_ and F_2:3_ populations are derived from the cross of two inbred rice parents, the segregating fertility genes sensitive to the environment are separated. Hence, traits related to seed setting rate, which are correlated between the two environments, do not show a significant correlation. However, the main correlations among environments are observed in PL, PBN, SBN, GN, and PSD, indicating that the characteristics of Chaoyou1000, such as large panicles and high panicle density, exhibit relatively high stability across different environments. It is noteworthy that the correlation coefficients for these traits were relatively small, underscoring the significant influence of the environment on rice panicle traits.

### 2.3. Construction of Genetic Map

The genetic map for the F_2_ population was created using 123 marker pairs and 142 rice plants. The physical location and primer information are provided in Appendix A. The total physical length of each chromosome is presented in Table 1, amounting to 373 Mb. The average distance between markers is 2.94 Mb, leading to a cumulative coverage of 2262.48 cM for the entire genome, with an average inter-marker distance of 17.81 cM. These conditions were established to facilitate the accurate detection of quantitative trait loci (QTLs).

### 2.4. Main-QTL Analysis

A total of 33 QTLs were discovered for the nine panicle-related traits, distributed across all chromosomes except for chromosome 11. These QTLs were located in 17 distinct regions. Specifically, 23 QTLs were identified in the F_2_ population, while 10 QTLs were identified in the F_2:3_ population. Notably, only two of these QTLs were found to be identical in both populations. The LOD scores for these QTLs ranged from 2.54 to 12.08, and each QTL accounted for a phenotypic variation ranging from 4.96% to 46.09% (Table 2).

Panicle number per plant (PN): In the F_2:3_ population, a single QTL named *qPN-7F* was detected for the trait of panicle number (PN). This QTL is positioned between the RM1132 and RM3555 markers on chromosome 7 and explains approximately 10.60% of the phenotypic variation. However, no QTL was identified for this trait in the F_2_ population.

Primary branch number (PBN): For the primary branch number (PBN) trait, two QTLs were identified, one in each population. These QTLs are named *qPBN-4L* and *qPBN-4F*. In the F_2_ population, *qPBN-4L* was detected, explaining 46.09% of the phenotypic variation. In the F_2:3_ population, *qPBN-4F* was identified and explained approximately 17.52% of the phenotypic variation.

Secondary branch number (SBN): In the F_2_ population, four QTLs were identified, and in the F_2:3_ population, one QTL was detected for the trait of secondary branch number (SBN). These QTLs were located on chromosomes 4, 7, 9, and 12, respectively. They accounted for the phenotypic variation at rates ranging from 7.26% to 16.75%.

Panicle length (PL): A total of five QTLs for panicle length (PL) were identified, and distributed across chromosomes 1, 5, 7, and 8. Two of these QTLs were detected in the F_2_ population, and three in the F_2:3_ population. These QTLs explained phenotypic variation rates ranging from 7.21% to 21.50%. Notably, QTLs *qPL-7L* and *qPL-7F* were identified in both populations. QTL *qPL-7L* accounted for 21.50% of the phenotypic variation, while *qPL-7F* explained 8.67% of the phenotypic variation.

Grain number per panicle (GN): Five QTLs associated with grain number per panicle (GN) were identified in both populations. These QTLs were located on chromosomes 2, 3, 4, and 7, and collectively explained phenotypic variation ranging from 9.22% to 26.64%.

Filled grain number per panicle (FGN): In the F_2_ population, three QTLs were successfully identified. Each of these QTLs accounted for a portion of the phenotypic variation, with their contributions ranging from 9.65% to 20.36%. Notably, two of these QTLs, *qFGN-3L* and *qFGN-4L*, explained more than 10% of the phenotypic variation.

Empty grain number per panicle (EGN): For the trait of empty grain number per panicle (EGN), four QTLs were identified on chromosomes 1 and 2 in both populations. These QTLs collectively explained phenotypic variation rates ranging from 9.25% to 29.98%.

Seed setting rate (SSR): In the F_2_ population, six QTLs for various traits were detected on different chromosomes, including chromosomes 1, 2, 5, 6, and 10. Each of these QTLs explained a portion of the phenotypic variation, with their contributions ranging from 4.96% to 30.86%.

Panicle setting density (PSD): Two QTLs were identified in both of the different populations, and these QTLs were located on chromosome 4. Each of these QTLs explained a portion of the phenotypic variation, with their contributions ranging from 13.82% to 15.28%.

### 2.5. Digenic Epistatic QTL Pairs (E-QTL) Underlying Panicle Morphology Traits

To comprehend the types of gene action governing nine quantitative traits in rice, we conducted quantitative trait locus (QTL) mapping to differentiate between the main-effect QTLs (M-QTLs) and digenic epistatic QTLs (E-QTLs) responsible for each trait. 50 pairs of digenic epistatic QTLs were detected in this study; 32 and 18 pairs each in the F_2_ and F_2:3_ populations. In the F_2_ population, 2, 1, 3, 1, 2, 2, 9, 8, and 4 digenic epistatic QTL pairs for PN, PL, PBN, SBN, GN, FGN, EGN, SSR, and PSD were detected (Table 3). Among them, 20 pairs occurred between one main QTL and one locus, 3 pairs between two main QTLs, and 9 pairs between two loci without a main effect. Sixteen E-QTLs improved panicle-related traits. Notably, one pair was detected for both PBN and GN, explaining 9.35% and 17.09% of phenotypic variances, respectively. Another pair was detected for both EGN and SSR, explaining 21.59% and 18.77% of phenotypic variances, respectively. In F_2:3_ population 5, 2, 3, 2, 3 and 3 E-QTL pairs for PN, PL, PBN, GN, SSR and PSD were identified (Table 3). Five pairs occurred between one main QTL and one locus, and the rest occurred between two loci without a main effect. Nine E-QTLs enhanced panicle morphology traits. No E-QTL was found to control two or more traits. One E-QTL for SSR was detected in both the F_2_ and F_2:3_ populations, explaining 18.77% and 30.75% of phenotypic variances, respectively.

### 2.6. Q × E Interaction Analysis in F_2_ and F_2:3_ Populations

Environmental factors can have a significant impact on panicle-related traits, ultimately influencing rice yield. In this study, 1, 3, 1, 4, 2, 1, 1, and 1 QTLs were identified to interact with environmental factors for PN, PL, PBN, SBN, GN, FGN, SSR, and PSD (Table 4). These QTLs contributed to the environmental interaction effects, with individual QTLs explaining a portion of the variation between 0.15% and 4.25%. Notably, *qPN-7F* located on chromosome 7 had the largest contribution rate at 4.25%. The environmental interaction contribution rates of *qPN-7F*, *qPL-5F*, and *qFGN-7L* exceeded their own contribution rates for additive effects. In contrast, the remaining 11 QTLs had environmental interaction contribution rates that were lower than their own contribution rates for additive effects. It is worth noting that *qPL-7F* and *qPBN-4F* exhibited insignificant interactions with the environment. The contributions from the QTLs involved in these interactions were significantly larger than those influenced solely by environmental factors.

## 3. Discussion

Rice is one of the world’s most important staple crops, feeding more than 50 percent of the Chinese population [20]. The characteristics of the rice panicle morphology traits play a pivotal role in influencing yield. While numerous panicle-related quantitative trait loci (QTL) and their associated genes have been mapped and cloned, there is a continuing need to identify additional QTLs and genes to elucidate the intricate mechanisms underpinning robust rice yields. It is worth noting that the majority of panicle-related traits and yield-related QTLs are typically identified through the use of DH, RIL, NIL, or F_2_ populations that are constructed from common rice varieties. There has been relatively limited exploration of QTLs using progeny populations derived from super hybrid rice varieties. In our research, we employed F_2_ and F_2:3_ populations derived from the high-yielding hybrid rice variety, Chaoyou 1000, developed through the hybridization of the sterile line Guangxiang 24S and the restorer line R900. Chaoyou 1000 has many beneficial traits for rice breeding, including the focal trait of increased grains per panicle, as well as panicle length, seed setting rate, and panicle setting density. Yield related traits are controlled by genetic components and are affected by the environment including temperature, daylength and nutrition. Nitrogen loss in the field could result in increased variability in the primary branch number, secondary branch number, and spikelet number per panicle [21]. The genetic components contributed to these traits are composed by gene effect and epistatic interaction [22]. In our research, the number of E-QTL exceeded that of M-QTL, with most E-QTL identified having two loci without main effects. Furthermore, the total phenotypic variance explained by E-QTL was similar or even greater than that of M-QTL for most traits. Consequently, we concluded that E-QTL has a significantly larger impact on panicle morphology traits than M-QTL.

In this study, a total of 33 QTLs were identified for the nine panicle-related traits, and these QTLs were distributed across all chromosomes except for chromosome 11. Including 1 QTL for PN, 2 for PBN, 4 for SBN, 5 for PL, 5 for GN, 3 for FGN, 4 for EGN, 6 for SSR, and 2 for PSD. These QTLs included 8 novel QTLs, and 25 QTLs may be located near previously described and cloned QTLs. According to previous studies, *qEGN-1L* and *qSSR-1L* should be in the same interval with the cloned *Gn1a* [12]. *qEGN-2L* and *qSSR-2.1L* are located near the cloned *EP3* [13]. *qEGN-2.2L* and *qSSR-2.2L* might be in the same location as the cloned *qSPP-2* [14]. *qGN3L* and *qFGN-3L* might be in the same location as the cloned *qSPP-3* [15]. The location of *qSSR-5L* and *qPL-8L* is nearly the location reported by Tao [23]. *qPN-7F* is nearly the location reported by Wang [24]. *qSBN-9L* should be in the same location reported by Liu [25].

Two QTL clusters were detected on the fourth and seventh chromosomes, respectively. QTLs associated with PBN, SBN, GN, FGN, and PSD under the two distinct environments were detected in the CM4-27~RM1113 interval on chromosome 4, including *qPBN-4L*, *qPBN-4F*, *qSBN-4.2F*, *qGN-4.1L*, *qGN-4.2F*, *qFGN-4L*, *qPSD-4.1L*, and *qPSD-4.2F*. QTLs related to PL, SBN, FGN, and GN under the two different environments were detected in the RM3775~RM11 interval on chromosome 7, including *qPL-7L*, *qPL-7F*, *qSBN-7L*, *qFGN-7L*, and *qGN-7L*. The two QTL clustering intervals detected were similar to those detected in previous studies. The QTL cluster on chromosome 4 was consistent with QTL *qGN 4-1*, *Spr3/OsLG1,* and *OsNAL1*. *qGN4-1* is a locus for grain number and was mapped overlapped with QTL cluster on chromosome 4 [16,26]. *Spr3/OsLG1* and *OsNAL1* were included in this region which has a great impact on plant height, PL, PBN, SBN, GN, FGN, EGN, and SSR [17]. *Spr3/OsLG1* is associated with domestication, while *OsNAL1* has been shown to contribute to yield divergence in cultivars. Hence, *OsNAL1* may represent a crucial selected locus in Chaoyou 1000 contributing to the achievement of a super high yield. The QTL cluster on chromosome 7 includes *Ghd7* detected by the previous study (Figure 3). QTLs control PBN, SBN and GN and are located in the same position interval. *Ghd7* simultaneously controls grain number per panicle, plant height and heading date. Using the F_2:3_ population and RIL constructed by Zhengxian 97 and Minghui 63, *Ghd7* is fine-mapped in a 2284 kb region on chromosome 7 [18].

In total, eight new QTLs were identified in the study, namely *qSBN-4.1L*, *qSBN-12L*, *qPL-1F*, *qPL-5F*, *qGN-2F*, *qEGN-2.3F*, *qSSR-6L,* and *qSSR-10L.* Of these, *qPL-1F*, *qPL-5F*, *qGN-2F*, *qSSR-6L,* and *qSSR-10L* exhibited phenotypic contributions exceeding 10%, suggesting that these QTLs may have a primary effect on the traits (Figure 4 and Table 2).

## 4. Materials and Methods

### 4.1. Plant Material

In this study, the sets of 142 plant F_2_ populations were developed from Guangxiang 24S (Super 900) and R900. From these F_2_ populations, 142 F_2:3_ family populations were derived. Guangxiang 24S is an *indica* low-temperature sensitive dual-use nuclear sterile line. On the other hand, R900 is an *indica* rice-restorer line. The crossbreeding between Guangxiang 24S and R900 resulted in the development of Chaoyou 1000, an excellent super high-yield hybrid rice variety, whose yield can exceed 15 t/ha^2^.

### 4.2. Field Experiment

The F_2_ population and two parents were grown at Lingshui county of China (18°20′ N, 110°00′ E) from November 2017 to April 2018, and the F_2:3_ population and two parents were planted in Hangzhou of China (119°54′ E, 30°04′ N) in the summer season of 2018. These two locations represent two types of ecology. The climate in Hangzhou is characterized by a subtropical monsoon climate, while Lingshui experiences a tropical monsoon climate. In Hangzhou, the minimum temperature during the growing season was 18.6 °C, reaching a maximum of 36.0 °C, with an average temperature of 26.10 °C. Monthly rainfall ranged from 157 mm to 211 mm, with an average humidity of 70.3%. In Lingshui, the minimum temperature during the growing season was 11.8 °C, reaching a maximum of 31.4 °C, and the average temperature was 23.17 °C. Monthly rainfall varied from 14 mm to 59 mm, with an average humidity of 73.0%. Hangzhou experiences a daily sunshine duration of 10 to 12 h during the growing season, while the daily sunshine duration in Lingshui during the growing season is generally between 8 to 10 h. (All meteorological data are available at https://tianqi.2345.com/, accessed on 21 September 2018). In addition, the Hangzhou field boasts fertile soil, while the soil in Hainan is sandy, leading to significant nutrient loss. All individuals were grown with a spacing of 20 cm. Two parents and each family were planted in six rows of six plants. Formulated fertilization: N 142.5 kg/hm^2^, P_2_O_5_ 60.0 kg/hm^2^, and K_2_O 142.5 kg/hm^2^. Nitrogen fertilizer is urea (N 46%), and phosphorus fertilizer is calcium magnesium phosphate fertilizer (P_2_O_5_ 16%) and potassium chloride (K_2_O, 60%). All the fertilizer was applied at once as the base fertilizer. The plots were arranged in a completely randomized block design with three replications. All field management followed local farmers’ practices.

### 4.3. Traits Investigation and Data Analysis

At about 25 days after flowering, the panicle number (PN) of parent lines and F_2_ and F_2:3_ populations were measured. Upon maturity, the main panicle was carefully chosen for the measurement of various panicle-related traits, including panicle length (PL, in centimeters), which was measured as the length from the panicle node to the tip of the panicle of five plants in a row plot, primary branch number (PBN), secondary branch number (SBN), grain number per panicle (GN), filled grain number per panicle (FGN), and empty grain number per panicle (EGN), which were measured by the main panicles of five plants in a row plot. Two derived traits, panicle setting density (PSD = GN/PL) and seed setting rate (SSR = FGN/GN), were calculated. For this purpose, all the main panicles of 15 random individuals’ parents and each of the F_2:3_ families were selected to measure their panicle morphology traits. In the case of the F_2_ population, all the main panicles of each individual were evaluated for panicle morphology traits. We conducted a *t*-test to compare the values of these nine traits among the parents. Additionally, skewness and kurtosis analyses in F_2_ and F_2:3_ populations were performed using Excel 2016 (16.0.17029.20028). The statistical description of parents, F_2_ and F_2:3_ populations, and the correlation between different traits and different populations were analyzed using the SAS 8.0 software.

### 4.4. DNA Extraction and Marker Analysis

Genomic DNA was extracted from young plant leaves using the CTAB method [27]. A total of 123 polymorphic markers were used to identify the genomic genotypes of the F_2_ population, of which 34 pairs of SSR and STS markers were from 242 public markers, the remaining 93 markers were screened by 373 pairs of indel markers developed by parent sequencing. PCR amplification was performed following the method described by Ren [28], and the PCR product was recorded after fragmentation analysis—capillary electrophoresis.

### 4.5. Genetic Map Construction and QTL Detection

The genetic linkage map is constructed by using Icimapping 4.1 software. QTL mapping was conducted by integrating the genetic map with trait data. To convert recombination frequencies to centimorgans, we employed the Kosambi mapping function. For QTL detection, the default LOD thresholds were set at 2.5, serving as the critical value for QTL identification. The percentage of variance explained by a single QTL and allele was estimated at the maximum likelihood QTL position, and its additive effect and dominant effect, genetic parameters and variation percentage were estimated. Its QTL naming followed the Khush system [29,30]. The digenic epistatic QTL analysis was performed using IciMapping Version 4 with the BIP functionality and ICIM-EPI method, following the manual instructions. The LOD threshold was set at the default value of 5.0. The QTL by environment interactions were analyzed using IciMapping Version 4, employing the Multiple Environment Trials (MET) functionality and ICIM-ADD method.

## 5. Conclusions

In our study, we investigated the rice panicle traits of PN, PL, PBN, SBN, GFN, FGN, EGN, PSD, and SSR in progeny populations derived from super hybrid rice across different environmental conditions. In this study, eight new QTLs were identified: *qSBN-4.1L*, *qSBN-12L*, *qPL-1F*, *qPL-5F*, *qGN-2F*, *qEGN-2.3F*, *qSSR-6L*, and *qSSR-10L*. Among these, *qPL-1F*, *qPL-5F*, *qGN-2F*, *qSSR-6L*, and *qSSR-10L* exhibited phenotypic contributions exceeding 10%, suggesting that these QTLs may have a primary effect on the traits. While many of the identified loci were consistent with findings from previous studies, the discovery of novel QTLs not only enhances our understanding of the genetic basis of panicle traits but also holds significant potential for application in the breeding of high-yield rice varieties.

## Figures and Tables

**Figure 1 plants-13-00179-f001:**
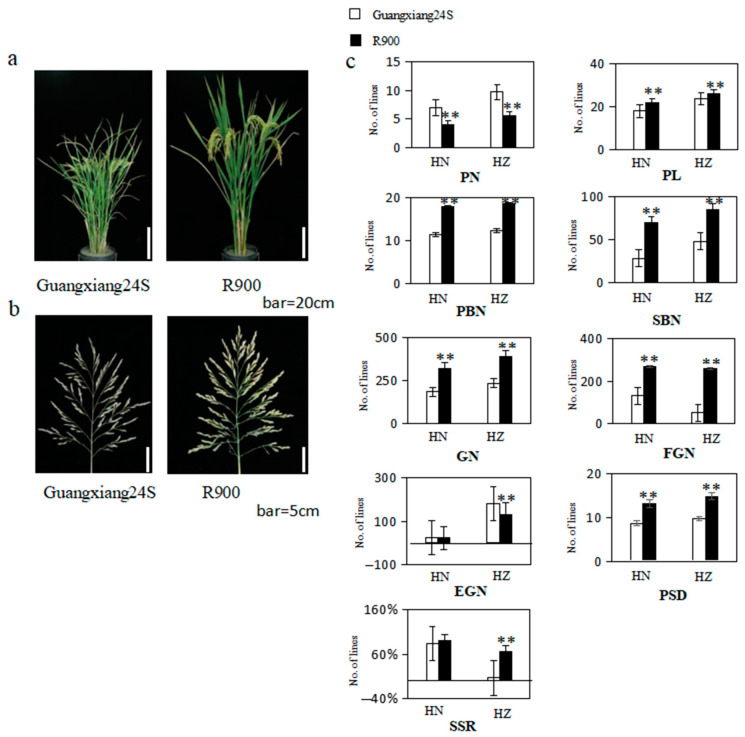
(**a**) Picture of the parents’ plants after the grout period, bar = 20 cm; (**b**) picture of the parents’ panicle after the grout period, bar = 5 cm; (**c**) the bar chart rice yield-related character of parents in Hainan (HN) and Hangzhou (HZ). PL: panicle length; PN: panicle number per plant; PBN: primary branch number; SBN: secondary branch number; GN: grain number per panicle; FGN: filled grain number per panicle; EGN: empty grain number per panicle; SSR: seed setting rate; and PSD: panicle setting density; ** represent significant levels at *p* ≤ 0.01.

**Figure 2 plants-13-00179-f002:**
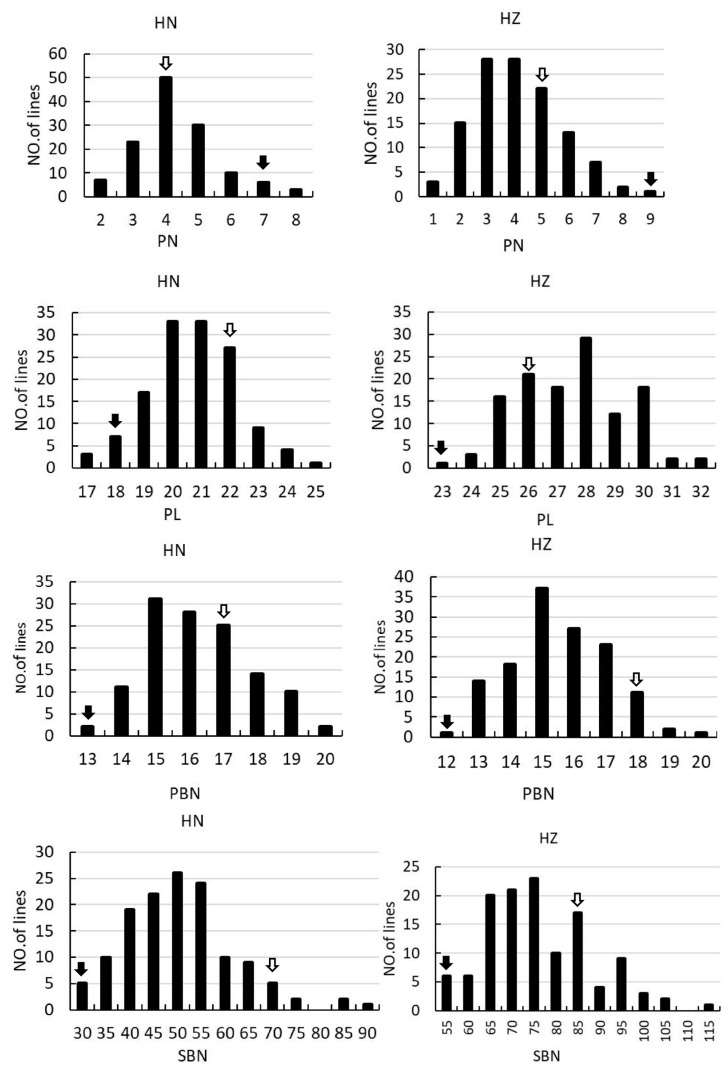
Frequency distribution of PN, PL, PBN, SBN, GN, FGN, EGN, SSR and PSD in the F_2_ and F_2:3_ population. The left columns indicate the rice grown in Hainan in 2017. The right columns indicate the rice grown in Hangzhou in 2018. The black arrows and the empty arrows indicate Guangxiang24S and R900, respectively.

**Figure 3 plants-13-00179-f003:**
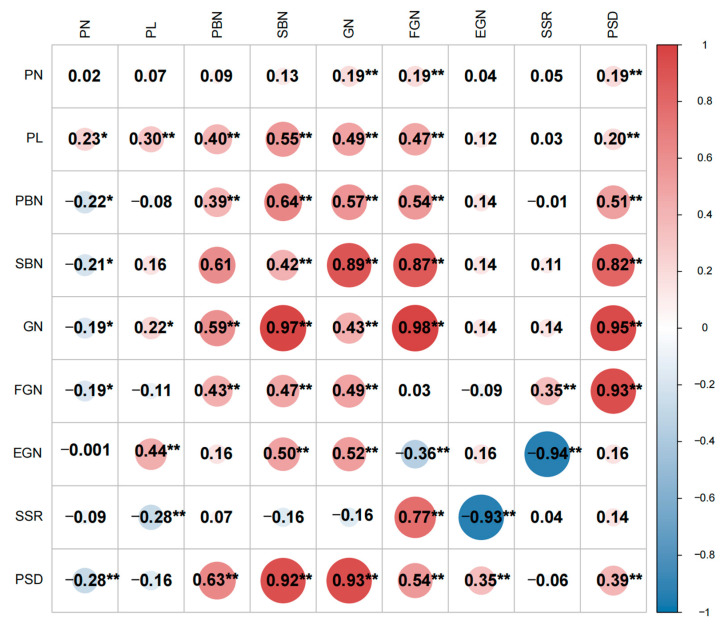
Correlation coefficients of yield-related traits in two population derived from Guangxiang 24S and R900. Data under and above the diagonal are correlation coefficients in F_2_ population and F_2:3_ population, respectively; the data on the diagonal are the coefficient of the same characteristic intercons in different environments; * and ** represent significant levels at *p* ≤ 0.05 and 0.01, respectively; trait PL: panicle length; PN: panicle number per plant; PBN: primary branch number; SBN: secondary branch number; GN: grain number per panicle; FGN: filled grain number per panicle; EGN: empty grain number per panicle; SSR: seed setting rate; and PSD: panicle setting density.

**Figure 4 plants-13-00179-f004:**
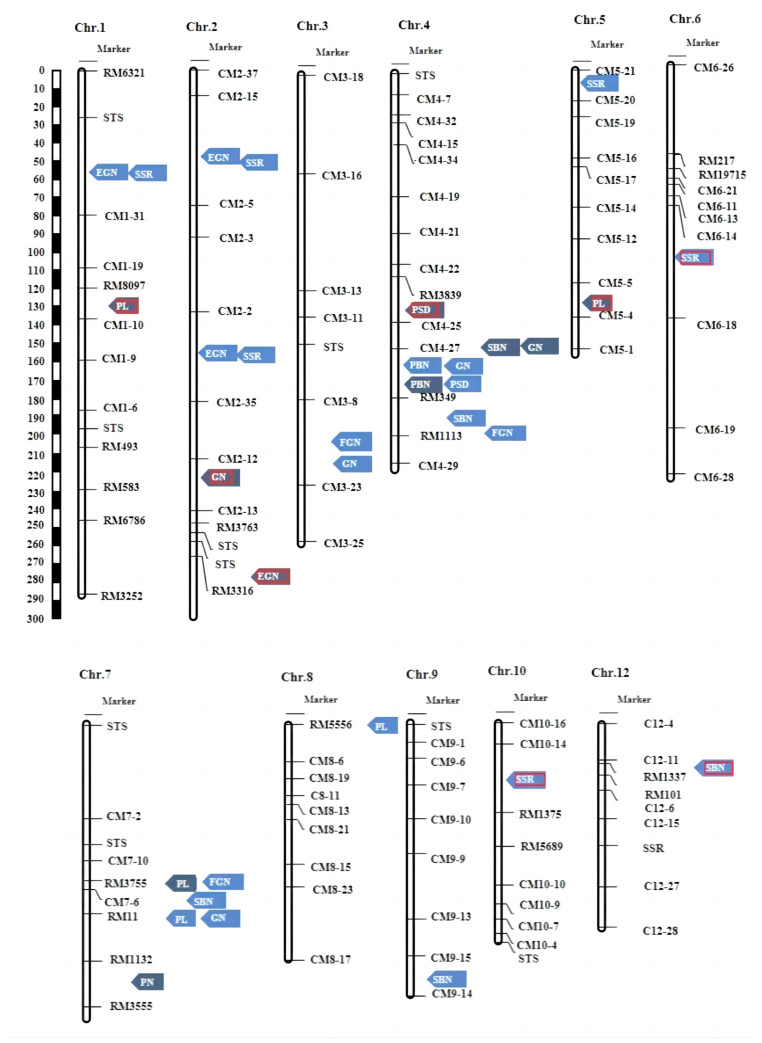
Locations of the main effect QTLs for PH, PN, PL, PBN, SBN, GFN, and FGN on the Linkage map; blue arrows indicate the F_2_ population’s main effect QTLs; grey arrows indicate the F_2:3_ population’s main effect QTLs. The red border represents the detected new QTLs.

**Table 1 plants-13-00179-t001:** Marker pairs distribution on 12 chromosomes.

Chr.	Physical Distance (Mb)	Mean Physical Distance (Mb)	Genetic Distance (cM)	Mean Genetic Distance (cM)	The Number of Marker
1	43.20	3.32	288.52	22.19	13
2	36.00	3.00	293.95	24.50	12
3	37.85	4.73	260.65	32.58	8
4	34.1	2.44	212.8	15.20	14
5	29.79	2.98	151.17	15.12	10
6	30.96	3.10	229.03	22.90	10
7	28.87	3.21	160.66	17.85	9
8	28.56	3.17	129.42	14.38	9
9	23.61	2.62	149.86	16.65	9
10	24.39	2.71	125.43	13.94	9
11	30.3	2.75	146.68	13.33	11
12	25.37	2.82	114.32	12.70	9
Total	373	3.07	2262.48	18.45	123

**Table 2 plants-13-00179-t002:** QTLs for yield-related traits in the F_2_ and F_2:3_ population of Guangxiang 24S and R900.

Trait Name	Environment	QTL	Chr.	Marker Interval	LOD	R^2^ (%)
PN	HZ	*qPN-7F*	7	RM1132-RM3555	2.83	10.60
PBN	HN	*qPBN-4L*	4	CM4-27-RM349	12.08	46.09
PBN	HZ	*qPBN-4F*	4	CM4-27-RM349	4.49	17.52
SBN	HN	*qSBN-4.1L*	4	RM349-RM1113	2.59	9.26
SBN	HZ	*qSBN-4.2F*	4	CM4-25-CM4-27	2.64	9.71
SBN	HN	*qSBN-7L*	7	CM7-6-RM11	4.25	11.80
SBN	HN	*qSBN-9L*	9	CM9-15-CM9-14	4.57	16.75
SBN	HN	*qSBN-12L*	12	RM1337-RM101	2.78	7.26
PL	HZ	*qPL-1F*	1	RM8097-CM1-10	2.56	11.97
PL	HZ	*qPL-5F*	5	CM5-5-CM5-4	2.65	10.28
PL	HN	*qPL-7L*	7	CM7-6-RM11	6.79	21.33
PL	HZ	*qPL-7F*	7	CM7-6-RM11	2.65	8.67
PL	HN	*qPL-8L*	8	RM5556-CM8-6	2.69	7.21
GN	HZ	*qGN-2F*	2	CM2-12-CM2-13	2.80	15.79
GN	HN	*qGN-3L*	3	CM3-8-CM3-23	3.28	26.64
GN	HN	*qGN-4.1L*	4	CM4-27-RM349	2.73	15.91
GN	HZ	*qGN-4.2F*	4	CM4-25-CM4-27	2.80	10.75
GN	HN	*qGN-7L*	7	CM7-6-RM11	2.54	9.22
FGN	HN	*qFGN-3L*	3	CM3-8-CM3-23	2.63	20.36
FGN	HN	*qFGN-4L*	4	RM349-RM1113	2.74	12.02
FGN	HN	*qFGN-7L*	7	RM3755-CM7-6	2.65	9.65
EGN	HN	*qEGN-1L*	1	STS-CM1-31	4.65	28.89
EGN	HN	*qEGN-2L*	2	CM2-15-CM2-5	4.85	28.24
EGN	HN	*qEGN-2.2L*	2	CM2-2-CM2-35	6.29	29.98
EGN	HZ	*qEGN-2.3F*	2	RM3316-CM2-6	3.03	9.25
SSR	HN	*qSSR-1L*	1	STS-CM1-31	3.23	24.54
SSR	HN	*qSSR-2.1L*	2	CM2-5-CM2-15	5.30	27.55
SSR	HN	*qSSR-2.2L*	2	CM2-2-CM2-35	2.84	22.01
SSR	HN	*qSSR-5L*	5	RM3616-CM5-17	2.89	4.96
SSR	HN	*qSSR-6L*	6	CM6-14-CM6-18	4.12	30.86
SSR	HN	*qSSR-10L*	10	CM10-14-RM1375	3.02	26.66
PSD	HN	*qPSD-4.1L*	4	CM4-27-RM349	2.73	13.82
PSD	HZ	*qPSD-4.2F*	4	RM3839-CM4-25	3.98	15.28

Trait: PL: panicle length; PN: panicle number per plant; PBN: primary branch number; SBN: secondary branch number; GN: grain number per panicle; FGN: filled grain number per panicle; EGN: empty grain number per panicle; SSR: seed setting rate; and PSD; panicle setting density. Environment: HZ: Hangzhou; and HN: Hainan. R^2^, phenotypic variation explained by the QTL. Gray shading for the detected new QTL.

**Table 3 plants-13-00179-t003:** Digenic epistatic QTL pairs (E-QTL) affecting panicle morphology traits in two different environment populations.

Population	Trait	Region1			Region2					
		Chr.	Marker Interval	M-QTL	Chr.	Marker Interval	M-QTL	LOD	AA	R^2^
F_2_	PN	2	CM2-15-CM2-5		8	CM8-15-CM8-23		5.90	−0.77	20.16
	PN	3	CM3-13-CM3-11		8	CM8-15-CM8-23		7.16	0.16	23.24
	PL	8	RM5556-CM8-6		12	CM12-15-SSR		5.51	0.43	10.47
	PBN	4	RM349-RM1113		11	CM11-21-CM11-41		6.80	−0.99	8.60
	PBN	2	CM2-2-CM2-35		4	RM349-RM1113		6.51	0.86	8.97
	PBN	4	CM4-27-RM349		8	CM8-13-CM8-21		5.73	0.32	9.35
	SBN	4	RM3839-CM4-25		9	CM9-13-CM9-15	*qSBN-9L*	5.65	0.53	14.89
	GN	4	CM4-27-RM349		8	CM8-13-CM8-21		6.87	−6.83	17.09
	GN	2	CM2-2-CM2-35		7	RM11-RM1132		6.29	1.78	20.90
	FGN	2	CM2-5-CM2-3		4	RM349-RM1113	*qFGN-4L*	6.13	37.80	13.52
	FGN	2	CM2-5-CM2-3		11	CM11-29-CM11-21		5.87	2.02	19.40
	EGN	2	CM2-15-CM2-5	*qEGN-2L*	2	CM2-15-CM2-5	*qEGN-2L*	11.68	−9.71	22.18
	EGN	1	STS-CM1-31	*qEGN-1L*	2	CM2-3-CM2-1		12.88	14.70	18.29
	EGN	1	STS-CM1-31	*qEGN-1L*	3	CM3-13-CM3-11		13.80	−2.45	21.78
	EGN	2	CM2-15-CM2-5	*qEGN-2L*	3	CM3-23-CM3-35		11.98	−9.40	22.85
	EGN	2	CM2-15-CM2-5	*qEGN-2L*	4	CM4-25-CM4-27		9.74	−8.90	17.66
	EGN	1	STS-CM1-31	*qEGN-1L*	5	CM5-4-CM5-1		11.37	-8.82	21.78
	EGN	2	CM2-15-CM2-5	*qEGN-2L*	5	CM5-4-CM5-1		10.15	6.75	16.95
	EGN	2	CM2-15-CM2-5	*qEGN-2L*	6	CM6-18-CM5-20		12.55	−4.77	21.59
	EGN	2	CM2-15-CM2-5	*qEGN-2L*	7	CM6-28-STS		12.12	−7.90	21.75
	SSR	1	CM6-28-STS		2	CM2-15-CM2-5	*qSSR-2.1L*	8.68	0.03	18.39
	SSR	2	CM2-15-CM2-5	*qSSR-2.1L*	5	RM3616-CM5-17		8.49	−0.05	17.83
	SSR	2	CM2-15-CM2-5	*qSSR-2.1L*	6	CM6-14-CM6-18	*qSSR-6L*	9.52	−0.03	18.77
	SSR	6	CM6-14-CM6-18	*qSSR-6L*	6	CM6-14-CM6-18	*qSSR-6L*	10.56	0.03	18.84
	SSR	1	STS-CM1-31	*qSSR-1L*	7	STS-CM7-2		9.57	−0.09	19.60
	SSR	1	STS-CM1-31	*qSSR-1L*	10	CM10-14-RM1375		8.54	-0.02	17.91
	SSR	2	CM2-15-CM2-5	*qSSR-2.1L*	10	CM10-14-RM1375		9.98	−0.04	18.62
	SSR	2	CM2-15-CM2-5	*qSSR-2.1L*	11	STS-CM11-5		7.37	−0.01	18.44
	PSD	2	CM2-3-CM2-1		4	CM4-27-RM349	*qPSD-4.1L*	5.08	2.01	11.82
	PSD	3	CM3-18-CM3-13		4	CM4-27-RM349	*qPSD-4.1L*	5.72	2.49	14.00
	PSD	4	CM4-27-RM349	*qPSD-4.1L*	7	RM11-RM1132		5.04	0.12	11.19
	PSD	4	CM4-27-RM349	*qPSD-4.1L*	8	CM8-13-CM8-21		7.51	−0.44	11.53
F_2:3_	PN	2	CM2-15-CM2-5		11	CM11-5-CM11-37		5.90	−0.25	30.52
	PN	4	CM4-19-CM4-21		12	SSR-CM12-27		5.75	−1.06	13.17
	PN	4	RM349-RM1113		7	RM11-RM1132	*qPN-7F*	7.68	-2.18	23.22
	PN	7	RM11-RM1132	*qPN-7F*	9	CM9-7-CM9-10		6.54	−0.93	24.74
	PN	10	CM10-14-RM1375		10	RM5689-CM10-10		6.09	0.44	19.83
	PL	9	CM8-17-STS		10	CM10-9-CM10-7		5.72	−1.31	8.76
	PL	4	CM4-25-CM4-27		7	CM6-28-STS		5.67	−0.90	15.36
	PBN	1	STS-CM1-31		8	RM5556-CM8-6		5.58	−0.43	23.29
	PBN	4	CM4-19-CM4-21		7	CM6-28-STS		5.91	0.82	16.80
	PBN	6	CM6-14-CM6-18		8	CM8-11-CM8-13		5.70	1.33	21.43
	SBN	4	CM4-27-RM349		8	CM8-15-CM8-23		6.73	−3.22	20.34
	SBN	4	CM4-15-CM4-34		9	CM9-13-CM9-15		5.82	13.56	15.50
	GN	2	CM2-15-CM2-5		9	CM9-9-CM9-7		5.96	1.26	28.02
	GN	4	RM3839-CM4-25	*qGFN-4.2F*	8	CM8-15-CM8-23		5.56	−0.82	21.04
	SSR	2	CM2-15-CM2-5		2	CM2-5-CM2-3		5.90	0.12	21.43
	SSR	2	CM2-15-CM2-5		5	CM5-19-RM3616		5.36	0.06	16.30
	SSR	4	CM4-25-CM4-27		5	CM5-19-RM3616		5.30	0.07	10.67
	SSR	2	CM2-15-CM2-5		6	CM6-14-CM6-18		7.59	0.13	30.75
	PSD	2	CM2-15-CM2-5		3	CM3-8-CM3-23		5.12	−0.63	25.82
	PSD	1	STS-CM1-31		4	CM4-25-CM4-27	*qPSD-4.2F*	5.60	−0.03	22.70
	PSD	4	CM4-25-CM4-27	*qPSD-4.2F*	9	CM9-13-CM9-15		6.45	−0.63	16.18

Trait: PL: panicle length; PN: panicle number per plant; PBN: primary branch number; SBN: secondary branch number; GN: grain number per panicle; FGN: filled grain number per panicle; EGN: empty grain number per panicle; SSR: seed setting rate; PSD; panicle setting density; QTL, quantitative trait locus; Chr., chromosome; M-QTL, main effect QTL listed in Table 3; AA, additive interaction effect; positive value represents E-QTL enhanced trait value, while negative value indicates E-QTL decreased traits value; and R^2^, phenotypic variation explained by the QTL.

**Table 4 plants-13-00179-t004:** M-QTL by environment interactions for panicle-related traits in F_2_ and F_2:3_ populations.

QTLs	Traits	Chr.	Marker Interval	R^2^ (%)	R^2^byA (%)	R^2^byE (%)	AbyE01	AbyE02
*qPN-7F*	PN	7	RM1132-RM3555	5.88	1.63	4.25	0.23	−0.23
*qPL-5F*	PL	5	CM5-5-CM5-4	5.45	1.24	4.21	0.15	−0.15
*qPL-7F*	PL	7	CM7-6-RM11	12.24	12.04	0.20	−0.10	0.10
*qPL-8L*	PL	8	RM5556-CM8-6	3.21	2.52	0.69	−0.17	0.17
*qPBN-4F*	PBN	4	CM4-27-RM349	22.16	20.91	1.25	−0.20	0.20
*qSBN-4.2F*	SBN	4	CM4-25-CM4-27	6.25	5.46	0.79	0.62	−0.62
*qSBN-7L*	SBN	7	CM7-6-RM11	6.01	5.27	0.74	−1.33	1.33
*qSBN-9L*	SBN	9	CM9-15-CM9-14	4.03	2.23	1.80	−2.11	2.11
*qSBN-12L*	SBN	12	RM1337-RM101	5.20	4.90	0.30	−0.42	0.42
*qGN-4L*	GN	4	CM4-27-RM349	7.47	6.98	0.49	−0.05	0.05
*qGN-7L*	GN	7	CM7-6-RM11	5.56	5.41	0.15	−2.09	2.09
*qFGN-7L*	FGN	7	RM3755-CM7-6	3.41	1.45	1.95	−10.03	10.03
*qSSR-5L*	SSR	5	RM3616-CM5-17	2.25	1.14	1.12	0.00	0.00
*qPSD-4.2F*	PSD	4	M45-M46	7.89	7.32	0.57	0.09	−0.09

Trait: PL: panicle length; PN: panicle number per plant; PBN: primary branch number; SBN: secondary branch number; GN: grain number per panicle; FGN: filled grain number per panicle; EGN: empty grain number per panicle; SSR: seed setting rate; PSD; panicle setting density; Chr., chromosome; R^2^, phenotypic variation explained by the QTL; AbyE01, environmental interaction effects in Hainan Lingshui; positive value represents QE’s enhanced trait value, while negative value indicates QE’s decreased traits value; AbyE02, environmental interaction effects in Hangzhou Fuyang; positive value represents QE’s enhanced trait value, while negative value indicates QE’s decreased traits value.

## Data Availability

The supporting data involved in this article are all original and can be provided upon request.

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
