# Peer review of "Genetic Dissection of Panicle Morphology Traits in Super High-Yield Hybrid Rice Chaoyou 1000"

_plants, 2024, doi:10.3390/plants13020179_

Round 1
Reviewer 1 Report
Comments and Suggestions for Authors
The introduction is compact, clearly written with relevant references. The objectives of this research are also clearly defined.
Line 9 - "The" - is not in bold
Line 45 - comma at the beginning of the paragraph
Line 63 + Figure 1 - different spelling of the GuangXiang 2AS rice variety
63-64 – it was appropriate to state what the abbreviations of phenotypic characters mean (here they are given for the first time in the text)
Line 82 - Table 1 could be included in supplementary data, Figure 2 is better suited to illustrate the nature of the characters which were analysed
Table 2 – it is quite interesting to present similar results in the form of a graphic "heat-map", the orientation in the graphic figure is significantly better than in the form of the sum of numbers
row 172 - in the legends of the table it is stated: yellow shading – in the table the shading is gray
Picture 3 - it could be a little bigger and above all better quality, the markers are hard to read.
Line 227 - which country? should be stated
The discussion is relatively short, it would be appropriate to discuss the results in more detail - in a similar structure to the results chapter
Line 263 – typical generation sequence designation is F2 and F3, F2 population and lines of F3 generation
Line 274 - two - capitalize
Line 293 – 5% agarose is a rather obscure separation system, fragmentation analysis – capillary electrophoresis would be more appropriate and accurate
Conclusions - it would be appropriate to better describe the conclusions and introduce new QTL.
Reviewer 2 Report
Comments and Suggestions for Authors
1) It is possible to detect genes by QTL analysis by preparing and measuring a large number of lines for genetic analysis. However, it is impossible to confirm that the genes detected in this way are valid genes without first confirming that the cultivation, experimental, and research methods are appropriate. If the cultivation and measurement methods are inappropriate, the QTLs detected from the data obtained from the genetic analysis of a large number of lines will be meaningless. This paper is extremely inadequate in describing the cultivation method, the environmental conditions under which the lines were cultivated, or the measurement method. It is impossible to evaluate whether the measurement data is correct or not. The above points should be properly described.
2) Line 9 The ears of rice are not spikes, but panicles. Please correct it.
3) In the abstract, it says that they were grown in two different environments, but the abstract should briefly describe how the two environments are different.
4) The panicle trait being investigated in this experiment is panicle morphology only. Therefore, it should be written that traits related to panicle morphology were investigated. Simply stating that the traits of the panicle were investigated could be misinterpreted as measuring physiological traits such as photosynthesis, respiration, translocation, enzyme activity, etc., which are highly relevant to yield.
5) The morphology of the rice panicle, especially the number of primary and secondary branches and the number of grains per panicle, is very strongly influenced by nitrogen nutrition. Therefore, the nitrogen content of the soil and the amount of fertilizer applied (fertilizer applied at each developmental stage) at the two locations where the rice was grown need to be properly documented. If there is a large difference in the amount of fertilizer applied and soil fertility between the two locations, it is impossible to clearly distinguish whether the differences in the morphological traits of the panicles are due to environmental factors or soil nitrogen and fertilizer application methods. If it is clear that nitrogen is equally effective at the time when young panicles are developing at these two points, then the difference in panicle morphological traits can be attributed to the environment at both locations. This must be clarified.
6) If the paper claims that the environmental differences between the two sites are meteorological differences, at least the average temperature, minimum temperature, maximum temperature, precipitation, solar radiation, and humidity (preferably expressed in vapor-pressure deficits) during the growing season at the two sites should be described. Without these meteorological observation data, readers will not be able to understand the specific differences between the two locations.
7) The method of investigation of the ear traits should be described in detail: describe how 15 individuals were selected from 36 plants (6 plants x 6 rows) (e.g., panilces with a medium number of tillers was selected). Were all 15 individuals surveyed or did you select the main culm or the longest panicle from each individual? Please describe exactly how you selected the panicles you examined.
8) Please describe the method used to measure panicle length. Different researchers use different methods to measure panicle length. Some researchers measure from the tip of the panicle to the base of the lowest growing (non-degenerated) primary branch. Others measure panicle length from the panicle node (small bulge) to the tip of the panicle, since the lower primary branch rachis-branch often stops growing and degenerates during the ear's growth. The latter is the official method of measurement, but there are some studies that use the former method. Please clearly state which method is used.
9) This expression is incomprehensible: "Lingshui is a semi-arid and semi-humid climate." Semi-arid and semi-humid are incompatible. Or, if the first half of the growing season is semi-arid and the second half is semi-humid, please write it as such.
10) Please describe in detail how you measure panicle setting density (PSD) and seed setting rate (SSR).
Reviewer 3 Report
Comments and Suggestions for Authors
I'm revising a manuscript on QTL mapping in rice for phenotypic traits related to panicle.
Within the work there are many errors of superficiality, method and conclusions. Starting with the title.
The methods, results and discussion extremely insufficient.
The comments below are reported considering the order section in the manuscript.
The marker number in the abstract is different from the one in line 289.
In the introduction a list of known genes linked to the phenotypic traits analysed in the work is given from line 44 to 46. no comparison with these genes was proposed, commented in the results and discussion.
The first paragraph of the results: phenotypic analysis of parents and population is extremely inadequate. the results are not described, but only tables and figures are referred to.
In the caption of figure 1 Plant Height is reported. Any analysis in relation to the traits is reported in the text. and in the population? what am i supposed to understand with the two bars reported in the figure?
Figure 2 is incomprehensible. I propose to break down by population/environment. the difference between the arrows should be described and commented on in the text.
Table 1 is completely unformatted and difficult to interpret. for all characters it is necessary to explain the phenotyping method and the unit of measurement in materials and methods.
Description of the correlation among traits/environment/population very unclear. Any comment on the positive and negative correlations between the traits. Is it expected to obtain these type of correlation?
Description of the genetic map insufficient, first give a summary table containing the number of marker , the density of marker and the length for each chromosome . .
Description of QTL. Table and text report different values for LOD and R2.
How can you have the same genetic interval for QTL reported in rows 134-137 if the LOD values and explained variability are so different? Table 3 is not complete. The peak marker, and the genetic position will be reported.
The paragraph E-related traits consists of 2 pages of table 4 and 7 lines of explanation. extremely inefficient. The method is not described in materials and methods.
The QxE paragraph in the results extremely insufficient. No description on methods in MM. Table 5 full of writing errors.
Paragraph 2.7 extremely insufficient in description. What are clusters? How do you demonstrate the new loci identified? where is the overlap explained if there is one with known genes and QTL?
In the discussion you talk about certain physical positions, e.g. line 247. what am i supposed to understand? the SPr3 gene is in what physical position? Where are located the QTL identified in this work?
Then the discussion is really inefficient in terms of content. It is repeated three times 1 and only 1 concept related to the populations used in the L linkage mapping.
Moving on materials and methods, also in this case the section is insufficient.
No description of the statistical analyses performed.
Heading date was evaluated? why not also report this trait and evaluate a possible correlation with the data under examination?
Description of the methods for developing genetic map is insufficient. Which map function was used to convert recombination frequencies to centimorgan? Ici mapping repeated twice line 295 and 297.
Description of methods for QTL mapping insufficient. Why was the threshold of the LOD on permutation test not calculated? How was the confidence interval calculated?
The part on E related traits, QxE and cluster description completely omitted.
Comments on the Quality of English LanguageEnglish must also be improved and corrected from countless superficial errors
Round 2
Reviewer 2 Report
Comments and Suggestions for Authors
The reviewer finds that appropriate revisions have been made to the paper as a whole. The reviewer notes a few problems.
1) Even though the cultivation in Lingshui was done during the winter season and in Hangzhou during the summer season, how can the temperature in Lingshui be more than 5°C lower than in Hangzhou, when a tropical monsoon climate is supposed to be more tropical and warmer than a subtropical monsoon climate? To begin with, there is a problem with the definition of climate, since a tropical climate refers to a climate with high temperatures and no winters throughout the year.
2) The number of primary rachis-branches, secondary rachis-branches, and spikelets on an panicle is greatly affected by nitrogen nutrition between the panicle initiation (PI) and the panicle booting stage. In an experiment in which many lines were tested with different anthesis and flowering periods, nitrogen nutrition during the PI period tended to be higher in lines that reached PI earlier because the PI period differed among lines when only basal fertilizer was applied, and the number of primary and secondary branches, and spikelets on each panicle tended to be higher (Wada and Sta. Cruz 1989. Japanese Journal of Crop Science 58: 732-739). If nitrogen is more easily lost in Lingshui soils, then this issue becomes even more important. Please add this point to the discussion.
Reviewer 3 Report
Comments and Suggestions for Authors
No comments
Comments on the Quality of English LanguageNo comments
Author Response
Dear Reviewer,
Thanks for your suggestion. We have thoroughly reviewed the manuscript and sought assistance from proficient native English speakers to enhance the language. Revised portions are marked in green on the paper.
Once again, we want to thank you for your valuable feedback and for taking the time to review our manuscript. Your input has helped to improve the quality of our work, and we appreciate your efforts in this regard.
Sincerely,
Jing Jiang